# Overcoming Open-Set Approaches to Adversarial Defense

**Edgar W. Jatho III**                                              *jatho@usna.edu*
*Department of Computer Science*
*United States Naval Academy*

**Armon C. Barton**                                          *armon.barton@nps.edu*
*Department of Computer Science*
*Naval Postgraduate School*

**Matthew Wright**                                         *matthew.wright@rit.edu*
*Department of Cybersecurity*
*Rochester Institute of Technology*

**Patrick McClure**                                       *patrick.mcclure@nps.edu*
*Department of Computer Science*
*Naval Postgraduate School*

**Reviewed on OpenReview:** *https://openreview.net/forum?id=iuQ9r8VSIX*

## Abstract

Machine learning (ML) models are increasingly proposed to replace or augment safety-critical information processing systems, yet their fragility to evasion attacks remains a well-documented, open problem. This work analyzes a class of deep neural network defenses that add a none-of-the-above (NOTA) class as an open-set-inspired, closed-set adversarial defense. We analyze seven prominent adversarial evasion attacks developed for computer vision classification and one attack developed for natural language processing classification, identifying how these attacks fail in the presence of a NOTA defense. We use this knowledge to adapt these attacks and provide empirical evidence that adding a NOTA class alone does not solve the core challenge of defending DNNs against evasion attacks. We release our adapted attack suite to enable more rigorous future evaluations of open-set-inspired defenses.

## 1 Introduction

Recent years have seen a steep increase in the number of successful applications of Deep Neural Networks (DNNs) across the sciences, industry, and business. This technology has enabled strides forward in areas as disparate as machine vision (Krizhevsky et al., 2012), neuroimaging analysis (McClure et al., 2019), astronomy (Valizadegan et al., 2022), cancer diagnosis (Savage, 2020), protein folding (Callaway, 2020), and natural language processing (NLP) (Brown et al., 2020). Despite these advances, efforts to leverage DNN technology in safety-critical systems have been hampered by the fact that current approaches create models that are highly susceptible to deception, particularly deception in the form of what are known as evasion attacks or adversarial examples. This is a well-documented, open problem that persists to today (Carlini, 2024).

To address this, many defense methods have been proposed to increase the robustness of DNNs to adversarial examples (Costa et al., 2024). The most widely implemented type of adversarial defense is adversarial training (Szegedy et al., 2013; Goodfellow et al., 2015; Madry et al., 2017). These defenses seek to create adversarial examples for a particular DNN from clean training examples. These adversarial examples are then labeled with the same label as the clean training examples used to generate them and are then used to train that DNN. This is done with the goal of making the DNN's predictions robust to the addition of adversarial noise

to the input. This robustness, however, may not defend against adversarial examples distant in input space from clean training examples (Costa et al., 2024).

One approach to addressing more distant adversarial attacks is to look to the open-set paradigm and create a None-of-the-Above (NOTA) class, a new class in addition to the existing classes in a particular dataset. In contrast to adversarial training, generated adversarial examples act as boundaries, are put into the NOTA class, and are used as training examples. The premise is that with carefully crafted NOTA augmentation examples, one can continuously seed the data-point-sparse space between data-dense regions of a classifier's input space throughout training and leverage the DNN to classify this vast space as NOTA. This approach is suggested as a potential remedy to the "inevitability" of adversarial examples under the standard DNN training paradigm (Shafahi et al., 2019). While covering all of the data-sparse regions of input space would be impractical, such approaches hold it might be sufficient to cover the regions that define the boundaries between and adjacent to classes and let the DNN generalize the remaining input space as NOTA.

Such defensive training strategies can be agnostic of the attack method or the neural network architecture. Instead, they seek to change the structure of how the input space is partitioned by the classifier through training with an additional NOTA class. The method is modular and can be added to any DNN classifier and may augment most present or future defense strategies. The entire approach has a fixed computational cost which only occurs during training, similar to adversarial training defenses. This makes it an attractive potential solution. Barton (2018) first demonstrated these open-set approaches to closed-set adversarial robustness and deep neural network defense with Boundary Padding (BP), a data augmentation approach which sought to insert a "padding class" between various classes in input space by linearly interpolating between two images and subsequently labeling the resulting example as a padding class (i.e., NOTA). This defense was successful against several standard evasion attacks.

However, evaluating open-set or "None-of-the-Above" adversarial defenses is held back by inadequate attack evaluations. A long line of work–from Athalye et al. (2018) through Tramer et al. (2020) to Suya et al. (2024)–shows that adversarial defenses judged "robust" often collapse once the attacker adapts attacks to a given defense. Once a researcher has conceived and built a defense, it is then incumbent on them to "switch hats," and apply their full knowledge and effort to break the specific defense they are proposing, by altering attacks as necessary (Carlini et al., 2019). Adapted attacks have been proposed for many defenses, but attacks adapted to defeat NOTA defenses have not been well-studied.

In this paper, we analyze seven prominent adversarial evasion attacks developed for computer vision classification and one attack developed for NLP classification, identifying how these attacks fail in the presence of a NOTA defense. We use this knowledge to adapt these attacks and provide empirical evidence that adding a NOTA class alone does not solve the core challenge of defending DNNs against evasion attacks. We release our adapted attack suite to enable more rigorous future evaluations of open-set-inspired defenses.

## 2 Background

Before analyzing NOTA-adapted attack strategies, it is necessary to discuss deep neural network models, evasion attacks, adversarial training, and NOTA defenses.

### 2.1 Deep Neural Network Classifiers

Adversarial attacks are most commonly executed against DNN models. In general, a DNN classifier can be described as a function $f(x) : \mathbb{R}^d \to \mathbb{R}^c$. Where the input is $x \in \mathbb{R}^d$, and the output is in $\mathbb{R}^c$, and is often called the logits. In the image classification domain, input $x$ is an $h * w * l$ pixel image such that $x \in [0, 1]^{hwl}$ and $c$ is the number of classes. In the text classification domain, an input text is a sequence of $n$ tokens, where each token is an element of $\{0, ..., v - 1\}$, where $v$ is the vocabulary size. Each token is mapped to an embedding vector $\mathbf{e}_i \in \mathbb{R}^m$, where $i$ is the token number and $m$ is the dimensionality of the embedding. The input, $x$, to the classification function, $f$, is then a sequence of $n$ embedding vectors. Classes are denoted by integer codes ranging from 0 to $c - 1$. The prediction of the DNN for an input $x$ is given by the equation $y = \text{argmax}(f(x))$.

## 2.2 Evasion Attacks

Evasion attacks imperceptibly modify the input to a model to produce a change in classification from the clean (i.e. originally intended) class to some other, untrue class (Chakraborty et al., 2018). This modified input is called an adversarial example. Adversarial examples are crafted such that given a classifier $f$, identifying $c$ distinct classes of objects $y_0, y_1...y_{c-1}$, and a clean input $x$, belonging to the class $y_{\text{true}}$, an input $x'$ can be crafted such that $\text{argmax}(f(x')) = y_{\text{pred}}$, where $y_{\text{pred}} \neq y_{\text{true}}$. This is accomplished by adding some perturbation $\delta_x$ to $x$ such that $x + \delta_x = x'$, where $\delta_x$ corresponds to only a small change in $x$ (Goodfellow et al., 2015). This small change can be defined differently for different input modalities and methods (Madry et al., 2017; Guo et al., 2021).

Our evasion attack threat model assumes the adversary's attack occurs after training and system deployment, (i.e., the adversary cannot manipulate training data). The adversary is assumed to be able to do one of two things. One, they can change the actual artifact in the real world. Examples of this approach would be donning anti-facial-recognition glasses to defeat identification systems (Sharif et al., 2016), applying an AI-camouflage pattern to a ship or tank to evade wide area motion imagery detection, or simply applying tape to precise positions on a stop sign to fool a self-driving car's image classification system into missing the sign (Eykholt et al., 2018). Two, an adversary with insider access to the data stream can change the direct input to the model by, for instance, altering pixels by imperceptible amounts to fool a classification system into misclassifying the pictured object (Goodfellow et al., 2015; Madry et al., 2017; Carlini & Wagner, 2017). In our threat model, the modifications to an input needed to change the predicted class of a model can be found using complete information of the classifier being attacked (i.e., a whitebox scenario).

## 2.3 Relevant Adversarial Defense Paradigms

Given the extensive literature that has accumulated regarding new methods to find effective, fast, and cheap evasion attacks, significant effort has likewise been expended pursuing effective measures to mitigate or eliminate these threats. Here we describe two DNN defense paradigms which, although near-complements of one another, have significantly different implications, strengths, and weaknesses. These include the well-known and ubiquitous *adversarial training* and the less well-exercised open-set approaches, like *NOTA-training* paradigms. The former's success at defending against existing state-of-the-art attacks is used as a baseline for evaluating the latter's robustness to original and adapted versions of the same attacks.

### 2.3.1 Adversarial Training Data Augmentation

The best known method for increasing DNN robustness remains adversarial training (Goodfellow et al., 2015; Szegedy et al., 2013). This is a data augmentation technique which adds adversarial examples to the true label class. The most common form, particularly for images, uses PGD, which, in many cases, leads to significantly improved adversarial robustness (Madry et al., 2017). Related gradient-based adversarial attacks have likewise been successfully employed to defend against adversarial attacks on NLP tasks.

Attacking to Training (A2T) is an adversarial training regime designed to defend LLM-based tools from adversarial attacks (Yoo & Qi, 2021). A2T introduces a gradient-based word-importance ranking derived from the gradient of the loss with respect to the embedding produced for each token in the input. It uses this basis to select the most impactful words to swap out and replace in order to achieve a successful attack (a change in classification). Once the attack is successful, the new example is added to the training dataset with a label that matches the original input's ground truth label. These examples are created on the fly during training as in other forms of adversarial training.

### 2.3.2 NOTA Defenses

In contrast to adversarial training, NOTA defenses generate adversarial examples or other data points in order to create training examples for the NOTA class. The premise is that adding carefully crafted NOTA augmentation examples to training updates can continuously sow the data-poor regions of input space with NOTA examples. This is particularly useful for using NOTA examples to separate data-rich regions of input

space which correspond to different classes. The DNN then learns to classify this space along with the adversarial examples which are perturbed into it, as NOTA.

Shafahi et al. (2019) showed mathematically that if a class occupies less than or equal to $1/2$ of the input space, then a random point from that class is, with high probability, either already misclassified or within a small $\epsilon$ distance of an input region classified as another class. They suggest that having a classifier assign a large portion of input space to an "I don't know" class could circumvent the "inevitability" of adversarial examples. In particular, surrounding regions belonging to standard classes with regions corresponding to NOTA could make adversarial attacks return NOTA-classified examples. Shafahi et al. (2019) doesn't guarantee that every attack will result in a NOTA example, but it does offer a theoretical rationale to support the idea that a NOTA buffer could absorb adversarial attacks.

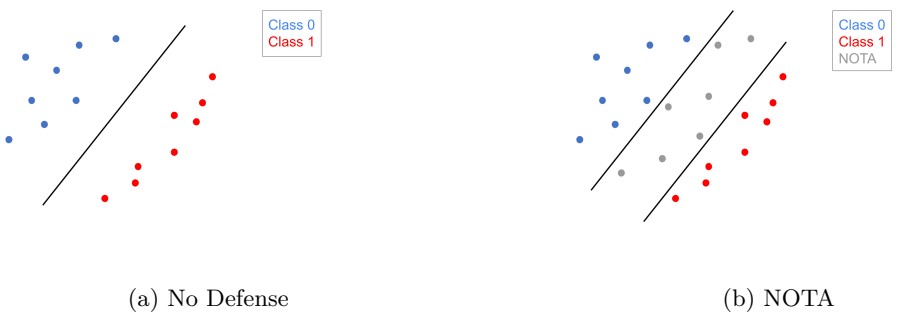

(a) No Defense               (b) NOTA

Figure 1: A conceptual illustration of a linear classifier for a binary classification task without (a) and with (b) a NOTA defense.

#### 2.3.2.1 NOTA, Open-Set, and Out-of-Distribution Methods

Although NOTA defenses leverage the open-set concept, their goal differs from most open-set constructs in the literature. The "I don't know" category is not generally used as a defense, but rather as a method to identify novel classes of data not already defined. Shao et al. (2020) characterize a research problem they call Open-Set Adversarial Defense (OSAD), where adversarial attacks are studied under open-set settings. In their framing, the goal is to both identify open-set samples (representing new classes) and defend against adversarial evasion attacks. They demonstrate that open-set classifiers were readily fooled using existing closed-set attack methods.

A related method is out-of-distribution defenses using thresholds, but this has been shown to be ineffective against simple adapted attacks. Enevoldsen et al. (2025) demonstrate that open-set recognition models that use thresholds of maximum softmax probability or maximum logit score to identify novel classes are also easily deceived using simple adaptations to existing adversarial attacks. Also, Grosse et al. (2018) show the ease with which adversarial attacks can achieve high confidence and low uncertainty adversarial examples which are misclassified by ML models, but not detected by an out-of-distribution threshold approach. Additionally, they demonstrate that such examples successfully transfer between different Bayesian models and approaches. Thus, their research implies that confidence and uncertainty alone cannot be used as a basis for defense against adversarial examples.

NOTA-type defenses are different in that they leverage the open-set concept to provide closed-set adversarial defense. NOTA defenses, therefore, do not facilitate or enable the identification of novel categories, nor do they use logit or uncertainty thresholds to identify adversarial examples. Rather, NOTA defenses leverage an additional none-of-the-above class to serve as the label for all adversarial examples, their derivatives, and open-set examples, relying on the DNN to generalize and identify adversarial examples as the NOTA class.

#### 2.3.2.2    Boundary Padding

BP was conceived after preceding research showed that various methods of creating NOTA class examples—such as using linear interpolation in the input space, or mixing methods in latent lower dimensional space using auto-encoders—showed promise at defending DNNs against adversarial examples produced by the CW attack suite using confidences of 20 or higher (Barton, 2018; Barton et al., 2021). However these methods struggled to perform against low-confidence Carlini Wagner $L_p$ attacks.

Another influence for BP resulted from Zhang et al. (2020), they introduce *Mixup*, an algorithm for instantiating linear behavior between training examples and increasing regularization and resistance to adversarial attack. Although BP discards the label-mixing aspect, it uses the simple mixing expression, $\lambda \cdot x_1 + (1 - \lambda) \cdot x_2$ for the image data as well as randomization of $\lambda$ in a new way to create its NOTA class examples. BP then is an attempt to more closely surround correctly labeled regions of input space with NOTA class examples. However, instead of "mixing" two separate training examples, as well as their labels, and training as that label combination, as is done in mixup, BP "mixes" a single training set example $x_1$ with an adversarial example, $x_1'$, derived using the PGD attack. The resulting BP image is labeled as the pure NOTA class and added to training on the fly. Note that no mixing of the labels occurs, all produced examples are instead labeled as the NOTA class and trained as such. This can lead to NOTA becoming a "padding" class that separates data from other classes (Figure 1).

In BP two variations of NOTA are produced on the fly and added to the training batch before batch-training commences, *mean BP* and *uniform BP*. In mean BP, $\lambda$ is set to 0.5 and limited gaussian noise is added to the resulting image. In uniform BP, $\lambda$ is set to a random number between 0.05 and 0.95 and a weighted average of the clean and adversarial example is performed. The resulting NOTA to clean data ratio is two to one, making it by far the largest class in the dataset. This large representation of NOTA vs. any other class reflects the intuition that, regardless of the number of finite classes that are defined, the vast majority of possible inputs in the input space do not correspond to any of the specified classes and, thus, should instead be assigned to the NOTA class.

## 3    Methods

### 3.1    Novel NOTA Defenses

In this paper, we not only evaluate NOTA-adapted attacks against BP, but also against two novel NOTA defenses, one that builds on BP and one that extends NOTA defenses to LLM-based classifiers.

### 3.2    Adversarial NOTA Envelopment

Given BP's existing loss formulations, as NOTA regions are planted and reinforced over the course of training epochs, it is likely that new NOTA examples will be planted in nearly the same input space locations epoch after epoch. Each new NOTA example then results in a steeper gradient to that portion of the input space, which in turn, will result in higher likelihood that future NOTA data augmentation will be created in close to the same place. This can have a reinforcing effect creating essentially a funnel or entrapment zone. As a result, NOTA regions clump near but not surrounding a class's partition space. Instead, the desired behavior of a NOTA defense should be to *surround* or envelop homogenously-classed datapoint-dense regions of input space with NOTA, while also populating data-sparse regions between them with NOTA.

Toward this end, the ANE defense retains all previous characteristics of BP, except it switches between two different losses in creating its NOTA data. One PGD loss maximizes the cross entropy (CE) with respect to the true label class as in BP. The second loss maximizes the cross entropy loss with respect to the NOTA class. This strategy is elegant in its simplicity, alternating between pushing away from the true label class to plant NOTA, and pushing away from existing NOTA partition to plant NOTA examples where they do not already exist. This leads to an attack loss of

$$\mathcal{L}(x, y_{\text{true}}; f) = \begin{cases} \text{CE}(y_{\text{true}}, \text{softmax}(f(x))) & \text{when } \beta \leq 0.5 \\ \text{CE}(y_{NOTA}, \text{softmax}(f(x))) & \text{otherwise,} \end{cases} \quad (1)$$

where $\beta \sim U(0,1)$. $\beta$ is used to randomly choose between pushing an adversarial example away from the true class and pushing the adversarial example away from the NOTA class.

### 3.3 NOTA A2T

To create NOTA examples for LLM-based models, we used a modified version of A2T (Yoo & Qi, 2021). Specifically, we labeled successful adversarial examples created by A2T (see Section 2.3.1) as the NOTA class, instead of the ground truth label. As in standard A2T, these examples are then used to augment the original training dataset during training.

### 3.4 NOTA-Adapted Attacks

In this paper, we analyze and adapt seven computer vision evasion attacks, CW $L_2$ and CW $L_\infty$ (Carlini & Wagner, 2017), AutoPGD with CE and AutoPGD with difference logits ratio (DLR) (Croce & Hein, 2020b), DeepFool (Moosavi-Dezfooli et al., 2016), Square Attack (Andriushchenko et al., 2020), and AutoAttack Croce & Hein (2020b), and one evasion attack for LLM-based classifiers, Gradient-based Distributional Attack (GBDA) (Guo et al., 2021).

#### 3.4.1 Adversarial Attack Success for NOTA-Defended Models

When attacking a NOTA-defended DNN classifier, $f$, with an input $x'$, an attack is successful only if the prediction is neither the true class, $y_{\text{true}}$, nor the NOTA class, $y_{NOTA}$. In other words, $\text{argmax}(f(x')) \neq y_{\text{true}} \wedge \text{argmax}(f(x')) \neq y_{NOTA}$. For every attack, it is important to evaluate whether the stopping criteria needs to be modified to use this NOTA-aware definition of a successful attack.

#### 3.4.2 Projected Gradient Descent (PGD) and AutoPGD (APGD)

Projected gradient descent is a straight-forward attack that leverages the same optimization that makes DNNs possible in the first place. In untargeted PGD, adversarial examples, $x'$ are discovered through gradient ascent and backpropagation (Madry et al., 2017). We let $\mathcal{L}(x, y; f)$ represent any loss function whose minimum results in $\text{argmax}(f(x)) = y_{\text{true}}$. Gradient ascent is then employed to iteratively update $x'$ such that the loss, $\mathcal{L}$ maximally increases and the resulting input leads to $\text{argmax}(f(x')) \neq y_{\text{true}}$.[1]

After each gradient update, $x'$ is projected to be within an $L_p$-bound, $\epsilon$, of $x$ and be in the set $[0,1]^{hwl}$, when dealing with images. The most common $L_p$-norms used in PGD and AutoPGD, and most other evasion attacks, are $L_2$ and $L_\infty$. $L_2(x, x')$ is the magnitude of the adversarial noise and $L_\infty(x, x')$ is the maximum of the absolute adversarial noise.

For $L_2$, $x'$ is updated using

$$x'_{t+1} = x'_t + \alpha \frac{\nabla_x \mathcal{L}(x', y; f))}{||\nabla_x \mathcal{L}(x', y; f))||_2}. \tag{2}$$

For $L_\infty$, $x'$ is updated using

$$x'_{t+1} = x'_t + \alpha \, \text{sign}(\nabla_x \mathcal{L}(x', y; f)). \tag{3}$$

Here, the gradient vector $\nabla_x \mathcal{L}(x', y; f)$ is the rate of change of the loss, $\mathcal{L}$, with respect to the input, $x$, and $\alpha$ is the learning rate. This procedure produces a perturbation for $x$ that pushes the DNN's prediction away from the true class, $y_{\text{true}}$. AutoPGD (Croce & Hein, 2020b) modifies the gradient-based update by using momentum and an adaptive learning rate.

In AutoPGD, two losses are commonly used: (1) CE and (2) DLR.

---

[1]This is equivalent to gradient decent on the negative loss.

#### 3.4.2.1 Cross Entropy (CE)

When performing untargeted PGD and AutoPGD with a CE loss, the probability of the true class, $y_{\text{true}}$, is minimized, as done in APGD-CE. The corresponding loss is

$$\mathcal{L}_{\text{CE}}(x', y_{\text{true}}; f) = \text{CE}(y_{\text{true}}, \text{softmax}(f(x'))). \tag{4}$$

When no NOTA defense is present, making any non-true class have the highest predicted probability by perturbing the input will lead to a successful adversarial example. However, this is not the case when a NOTA class is present. Making NOTA the highest probability class does not result in a successful adversarial example. We investigated using a linear combination of the cross entropy loss for the true class and the NOTA class in order to create an example that is not classified as either the true or NOTA class. However, we found that just maximizing the CE loss for the NOTA class led to a stronger attack, which we call Anti-NOTA (AN). This results in the AN loss being

$$\mathcal{L}_{\text{AN}}(x', y_{\text{NOTA}}; f) = \text{CE}(y_{NOTA}, \text{softmax}(f(x'))). \tag{5}$$

#### 3.4.2.2 Difference Logits Ratio (DLR)

CE is a shift-invariant loss, meaning the order of logits does not alter the output. Croce & Hein (2020b) introduces DLR, which is both shift and scale invariant. Scale invariance implies that rescaling the logits by a non-zero constant will not change the loss value. This ensures that the learning process—and by extension attacks seeking adversarial examples using gradient descent—will not be sensitive to or affected by the scale of the logits.

The DLR loss is defined as:

$$\mathcal{L}_{\text{DLR}}(x', y_{\text{true}}; f) = -\frac{f_{y_{\text{true}}}(x') - \max\limits_{i \neq y_{\text{true}}} f_i(x')}{f_{\pi_1}(x') - f_{\pi_3}(x')}, \tag{6}$$

where the elements of $f(x')$ are the logits output by the model for $x'$, $f_{y_{\text{true}}}(x')$ is the logit of the true class, and $\pi$ is the ordering of the components of $f(x')$ in decreasing order. DLR has been reported to be sometimes better-performing than CE with respect to attack success and more stable than Carlini-Wagner loss (Croce & Hein, 2020b).

When attacking a NOTA-defended model, DLR runs into a "target selection" issue. That is, the loss can actively incentivize making the logit for the NOTA class the largest logit. This occurs when the NOTA logit is the second-highest logit, after the true class logit. In the DLR loss, this issue arises when $\max_{i \neq y_{\text{true}}} f_i(x') = f_{\text{NOTA}}(x')$. Making NOTA the most likely predicted class can be problematic given the definition of attack success in the presence of NOTA. To address these issues, we modified the DLR loss to

$$\mathcal{L}_{\text{NOTA-DLR}}(x', y_{\text{true}}; f) = -\frac{f_{y_{\text{true}}}(x') - \max\limits_{\substack{i \neq y_{\text{true}} \\ i \neq y_{\text{NOTA}}}} f_i(x')}{f_{\pi_1}(x') - f_{\pi_3}(x')}. \tag{7}$$

This adaptation seeks to prevent the loss from seeking to maximize the logit of the NOTA class.

#### 3.4.3 Carlini-Wagner Attacks

Carlini-Wagner (CW) attacks (Carlini & Wagner, 2017) are a suite of attacks that use optimization methods to find successful adversarial examples that are created using the smallest possible perturbations, according to some metric $D(x, x')$. The two most common CW attacks use $L_2$ or $L_\infty$ metrics. CW attacks use the optimization loss of

$$\mathcal{L}_{CW}(x, x', y_{\text{true}}; f) = D(x, x') + \lambda_c \cdot \ell_{\text{CW}}(x', y_{\text{true}}; f) \tag{8}$$

The CW constraint term, $\ell_{\text{CW}}$, for untargeted attacks often takes the form of the margin loss

$$\ell_{\text{CW}}(x', y_{\text{true}}; f) = \max\left( f_{y_{\text{true}}}(x') - \max_{i \neq y_{\text{true}}} f_i(x') + \kappa, 0 \right), \tag{9}$$

where $\kappa$ is the "confidence" and controls how much the highest non-true logit, $\max_{i \neq y_{\text{true}}} f_i(x')$, exceeds the logit for the true class, $f_{y_{\text{true}}}(x')$. As in PGD, Carlini & Wagner (2017) add a box constraint that ensures $x' \in [0, 1]^{hwl}$ for images. The constant $\lambda_c$ is obtained through binary search and is used to increment or decrement the weight of the margin loss.

Like DLR, CW attacks suffer from a "target selection" issue when $\max_{i \neq y_{\text{true}}} f_i(x') = f_{\text{NOTA}}(x')$. To address this, the margin loss was adapted to

$$\ell_{\text{NOTA-CW}}(x', y_{\text{true}}; f) = \max\left( f_{y_{\text{true}}}(x') - \max_{\substack{i \neq y_{\text{true}} \\ i \neq y_{\text{NOTA}}}} f_i(x') + \kappa, 0 \right). \tag{10}$$

This prevents the loss from incentivizing increasing the NOTA logit.

### 3.4.4 DeepFool

---

**Algorithm 1:** DeepFool: Multi-Class (Moosavi-Dezfooli et al., 2016)

---

**1 input:**    Image $x$, classifier $f$ ;

**2 output:**    Perturbation $\hat{\delta}$ ;

**3** $y_{\text{pred}}(x) = \text{argmax}(f(x))$ ;

**4** Initialize $x'_0 \leftarrow x, i \leftarrow 0$;

**5 while** $y_{pred}(x'_i) = y_{pred}(x'_0)$ **do**

**6**   **for** $y \neq y_{pred}(x'_0)$ **do**

**7**     $w'_y \leftarrow \nabla f_y(x'_i) - \nabla f_{y_{\text{pred}}(x_0)}(x'_i)$;

**8**     $f'_y \leftarrow f_y(x'_i) - f_{y_{\text{pred}}(x'_0)}(x'_i)$;

**9**   $\hat{l} \leftarrow \text{argmin}_{y \neq y_{\text{pred}}(x'_0)} \dfrac{|f'_y|}{||w'_y||_2}$;

**10**   $\delta_i \leftarrow \dfrac{|f'_{\hat{l}}|}{||w'_{\hat{l}}||_2^2} w'_{\hat{l}}$;

**11**   $x'_{i+1} \leftarrow x'_i + \delta_i$;

**12**   $i \leftarrow i + 1$;

**13 return:**    $\hat{\delta} = \sum_i \delta_i$

---

The DeepFool attack (Moosavi-Dezfooli et al., 2016) can be seen as a gradient descent approach that sets the learning rate using the estimated minimum $L_2$ distance to a desired decision boundary. In the linear case, this method is optimal. However, that is not guaranteed for non-linear models. In practice, DeepFool often generates successful adversarial examples. The pseudocode is shown in Algorithm 1.

Two modifications are made to DeepFool to adapt it for NOTA, changing the stopping criteria to account for NOTA and changing the target selection to not select NOTA. To adapt the stopping criteria, the logical statement in Line 5 of Algorithm 1 is modified to $y_{\text{pred}}(x'_i) = y_{\text{pred}}('x_0) \vee y_{\text{pred}}(x'_i) = y_{\text{NOTA}}$. This means that the attack will continue until the model classifies the adversarial example as a class that is neither the initially predicted class nor the NOTA class, unless a maximum number of iterations is used. To adapt the

target selection, the logical statement in Line 6 of Algorithm 1 is changed to $y \neq y_{\text{pred}}(x_i') \wedge y \neq y_{\text{NOTA}}$. This prevents DeepFool from intentionally creating adversarial noise that would drive the input into a NOTA region.

### 3.4.5 Square Attack

Square attack is a score-based black-box $L_2$ and $L_\infty$ adversarial attack that does not use local gradient information and thus is immune to gradient masking (Andriushchenko et al., 2020). It uses a randomized search scheme and perturbations are introduced such that they lie on the boundary of the $\epsilon$ $L_2$-hypersphere or $\epsilon$ $L_\infty$-hypercube before $x'$ is projected back inside the box constraint (i.e., $x' \in [0, 1]^{lwh}$) for images. First, a side-length for the square that will be perturbed is chosen, according to a decreasing schedule. Next, a $\delta$ is chosen. If, on applying the $\delta$, the loss decreases, it is accepted. If not, it is rejected. If the new image classifies in a non-true class, the image is accepted. If not, the algorithm continues, repeating until it is either successful or the max number of iterations has been completed. Square attack seeks to minimize

$$\mathcal{L}_{SA}(x', y_{\text{true}}; f) = f_{y_{\text{true}}}(x') - \max_{i \neq y_{\text{true}}} f_i(x') \tag{11}$$

We adapted the attack by first preventing the NOTA class from being chosen as the initially correct label in the event the true labels were not provided. We altered the stopping criteria so that the attack was only successful if it resulted in a classification other than the true class and the NOTA class. We investigated modifying the loss to not incentivize increasing the NOTA logit, but, counterintuitively, found that doing so slightly weakened the attack.

### 3.4.6 AutoAttack

AutoAttack (Croce & Hein, 2020b) is considered the state-of-the-art adversarial evasion attack for computer vision. It is a highly effective ensemble of parameter-free attacks, combining the CE version of AutoPGD, the DLR version of AutoPGD, SquareAttack and the Fast Adaptive Boundary Attack (FAB) (Croce & Hein, 2020a). These separate attacks are used in sequence and until $\text{argmax}(f(x')) \neq y_{\text{true}}$. The adversarial robustness (ART) library (Nicolae et al., 2018), which we use in our computer vision evaluations, substitutes the DeepFool attack in for the FAB Attack. FAB is very similar to DeepFool, except that it uses additional iterative procedures to generate adversarial examples closer the decision boundary. However, a weakness of untargeted FAB is its extensive computational cost as dataset complexity and the number of classes increase (Croce & Hein, 2020a). For ease of testing and standardization, the hyperparameters for each attack in AutoAttack are constant across models, datasets, and measurement norms. Given its extensive strength and effectiveness, untargeted AutoAttack is a standardized benchmark used by the adversarial robustness community to compare model defense robustness to adversarial evasion attacks (Croce et al., 2020).

As AutoAttack leverages four separate subordinate attacks, it is necessary to ensure that the adapted NOTA version of AutoAttack calls the most effective adapted versions of each of these subordinate attacks. Additionally, the stopping criteria evaluated after each subordinate attack must be adjusted to exclude the prediction of NOTA as a successful outcome for the attack. We test both a NOTA-aware version of AutoAttack with NOTA-Aware APGD-CE and a NOTA-aware version of AutoAttack that replaces APGD-CE with APGD-AN.

### 3.4.7 Gradient-Based Distributional Attack

GBDA is an adversarial evasion attack that targets transformers used in LLM-based text classification, seeking to produce fluent and semantically similar adversarial examples (Guo et al., 2021). A major challenge in this approach is overcoming the non-differentiable nature of the mapping from the input text token numbers to embeddings vectors. GBDA seeks to address this challenge by using the Gumbel softmax to create a differentiable method for linking token numbers with an embedding.

The GBDA loss function is a linear combination of three components: (1) a CW margin loss (Equation 9), (2) a fluency constraint that preserves natural language flow, and (3) a semantic similarity constraint that

preserves meaning. This results in GBDA minimizing

$$\mathcal{L}_{\text{GBDA}}(\Theta, y, f) = \mathbb{E}_{\bar{\pi} \sim P_\Theta} \Big[ \underbrace{\ell_{\text{CW}}(e(\bar{\pi}), y_{\text{true}}; f)}_{\text{margin loss}} + \underbrace{\lambda_{lm} \text{NLL}_g(\bar{\pi})}_{\text{fluency}} + \underbrace{\lambda_{sim} \rho_g(x, \bar{\pi})}_{\text{semantic similarity}} \Big] . \tag{12}$$

In GBDA's objective function, $P_\Theta$ is the learned adversarial Gumbel-softmax distribution for a particular input. $\bar{\pi}$ is a sampled probability distribution from $P_\Theta$ over the vocabulary for each token in the input. $e$ is a function that computes the expected embedding vectors according to $\bar{\pi}$. $\text{NLL}_g(\bar{\pi})$ is the perplexity (an analog of fluency) of the expected embeddings according to the language model $g$, which in this case is GPT-2 (Radford et al., 2019). $\rho_g$ is the BERTScore (Zhang et al., 2019) (semantic similarity term) computed using $g$. $\lambda_{\text{lm}}, \lambda_{\text{sim}} > 0$, are hyper-parameters that adjust fluency and semantic similarity with attack strength. At each gradient-based update step, the attack draws multiple soft samples, $\bar{\pi}$, from $P_\Theta$, calculates the loss $\mathcal{L}_{\text{GBDA}}$, and then calculates the gradients to update $\Theta$. After $\Theta$ has been learned, the attack samples text inputs using $P_\Theta$ until a successful adversarial text is generated or the maximum number of samples has been reached.

Two modifications were made to adapt GBDA for NOTA, modifying the margin loss target selection and changing the stopping criteria when sampling potential adversarial text examples. As in CW attacks, the NOTA class must be eliminated from consideration when selecting a target for the margin loss. This was accomplished by setting the margin loss to $\ell_{\text{NOTA-CW}}(e(\bar{\pi}), y_{\text{true}}; f)$, where $\ell_{\text{NOTA-CW}}$ is defined in Equation 10. After learning the adversarial Gumbel softmax distribution for an input example, GBDA samples from this distribution until a successful adversarial example is found or the maximum number of samples is reached. In the adapted attack, the stopping criterion for this procedure was modified so that a successful adversarial example is defined to be one that is both not classified as the ground truth class and not classified as NOTA.

## 4 Experiments and Results

In this section, we discuss the evaluation of NOTA-adapted attacks against NOTA-defended models in both the computer vision and natural language processing domains. In the computer vision domain, we compare the effectiveness of adapted versions of seven benchmark evasion attacks against NOTA-defended models to the effectiveness of the original attacks against NOTA-defended models, adversarial-trained models, and undefended models. In the NLP domain, we compare the effectiveness of a NOTA-adapted version of a prominent NLP classification attack, GBDA, to the effectiveness of the original GBDA attack against both undefended and NOTA-defend LLM-based classifiers.

### 4.1 Experimental Setup

#### 4.1.1 CIFAR-10 and CIFAR-100

We performed computer vision experiments using the CIFAR-10 and CIFAR-100 datasets. Our datasets are split before training such that 4% of the former training set are quarantined as a validation set to enable early-stopping model selection, based on a combination of best validation accuracy and best validation adversarial robustness. In very close models, we slightly favored best validation accuracy over validation adversarial robustness. The test set is strictly reserved for testing a specific model that has been previously selected using only ASR and accuracy performance from the validation set. In testing, accuracy was calculated from the full test set.

For CIFAR-10 and CIFAR-100, we used standard Wide Residual Networks (WRNs) with dropout and batch normalization, configured as suggested by Zagoruyko & Komodakis (2016). Dropout is set to 30% drop probability during training. For all models, we used a WRN-12-6 (i.e. 12-unit deep and 6-unit wide WRN), a common setup for these datasets. For each model, we used an ADAM (adaptive moment estimation) optimizer with default settings, $b_1 = 0.9$ and $b_2 = 0.999$ (Kingma & Ba, 2017) with sharpness-aware minimization (SAM) (Foret et al., 2021). ADAM assisted the model in efficiently converging on a solution, whereas SAM, by seeking out minima of the training loss landscape and minimizing loss curvature, smooths the boundaries between partitions expressed in input space. When performing mini-batch gradient descent training, two separate batches of 32 were drawn from the training set for each training cycle. The first used standard

dataset augmentation, a random up to 10% shift up or down, and left or right, as well as random horizontal flipping and a random, up to a 15 degree rotation clockwise or counter clockwise. The second batch was a "clean" batch drawn from a separate iterator without data augmentation. Each batch was used both as a benign training batch and also to create NOTA examples based on each training example.

Every 150 batches during training, the same 30 examples were used (previously separated from the validation set) to create 30 untargeted, zero-confidence CW $L_2$ adversarial examples with a maximum of 10 iterations. These adversarial examples were then used to calculate a validation attack success rate (ASR), which is used in model selection. ASR was calculated by determining the number of adversarial examples that successfully drove the model to classify the image as a class other than the NOTA class or the true label. The process detailed here was the same for all model training and model selection, whether NOTA, adversarial training, or undefended models.

Table 1: Clean Model Test Accuracies

| CIFAR-10 | |
|---|---|
| **Model** | **Accuracy** |
| No Defense | 92.31% |
| Adversarial Training | 90.71% |
| Boundary Padding | 90.93% |
| Adversarial NOTA Envelopment | 92.20% |
| **CIFAR-100** | |
| No Defense | 70.53% |
| Adversarial Training | 66.47% |
| Boundary Padding | 68.13% |
| Adversarial NOTA Envelopment | 69.89% |

All models were WRN-12-6, with dropout of 30% and batch normalization.

Test set ASR is calculated on the selected model using adversarial examples created using sufficient test set samples to ensure reasonably small confidence intervals. We state our findings along with their 95% binomial confidence intervals. We test against CW $L_2$, CW $L_\infty$, AutoPGD-CE, AutoPGD-DLR, DeepFool, Square Attack, and AutoAttack. We adapt each of these attacks to counter the NOTA defense, and add an additional variant of AutoPGD, AutoPGD-AN. In parameterized attacks, we set max iterations to 100, i.e., CW $L_p$ attacks, square attack and DeepFool ($L_2$). All AutoPGD attacks, Square Attack, and AutoAttack are performed in both $L_2$ with max epsilon of 0.5 (maximum distance between $x$ and $x'$ by specified $L_p$ metric) and $L_\infty$ with max epsilon of 8/255, the distances specified for each by Robust Bench (Croce et al., 2020). Finally, DeepFool is tested in default $L_2$ with the standard max $\epsilon = 0.5$.

### 4.1.2 IMDB

We performed NLP experiments using the IMDB binary sentiment classification dataset (Maas et al., 2011). We used the same training and test sets as in Guo et al. (2021). We trained and evaluated a model that consists of BERT (Devlin et al., 2019) to create text embeddings, followed by a dropout layer with $p = 0.1$, a linear layer, and, finally, a softmax operation, as done in Morris et al. (2020); Guo et al. (2021). We fine-tuned this model for 5 epochs uisng AdamW (Loshchilov & Hutter, 2017) with a batch size of 16, a learning rate of 2e-05, a weight decay of 0.01, and a maximum sequence length of 512.

The NOTA defense and the attack evaluated also required several hyperparameters to be set. When creating NOTA examples during training, we used similar hyperparameters to those used for adversarial training by Geisler et al. (2024). Namely, we started generating adversarial examples for the NOTA class after the model had been trained for one epoch and created a number of successful adversarial examples equal to 20% of the training dataset each epoch afterwards. For original and NOTA-adapted GBDA, we closely followed the hyperparameters used by Guo et al. (2021). Specifically, we used a learning rate of 0.3, a CW confidence of 5, a perplexity coefficient of 1, a similarity coefficient of 20, and 100 Gumbel samples.

### 4.2 Results

### 4.2.1 CIFAR-10 and CIFAR-100

In Tables 2 and 3, attack success rates (ASR) are reported for original attacks against both undefended and defended models (adversarial training models, boundary padding models, and adversarial NOTA envelopment models, respectively).

Table 2: Attack Success Rates (ASRs), with 95% binomial confidence intervals, for Original and NOTA Attacks for CIFAR-10.

| Models | C&W Suite | | APGD CE | | APGD AN | | APGD DLR | | Square Attk | | DF | AA, Untrgtd | | AA-AN, Untrgtd | |
|---|---|---|---|---|---|---|---|---|---|---|---|---|---|---|---|
| | $L_2$, Conf:0 | $L_\infty$, Conf:0 | $L_2$, $\epsilon$:0.5 | $L_\infty$, $\epsilon$:0.031 | $L_2$, $\epsilon$:0.5 | $L_\infty$, $\epsilon$:0.031 | $L_2$, $\epsilon$:0.5 | $L_\infty$, $\epsilon$:0.031 | $L_2$, $\epsilon$:0.5 | $L_\infty$, $\epsilon$:0.031 | $L_2$, $\epsilon$:0.5 | $L_2$, $\epsilon$:0.5 | $L_\infty$, $\epsilon$:0.031 | $L_2$, $\epsilon$:0.5 | $L_\infty$, $\epsilon$:0.031 |
| No Defense vs. Orig. Attacks | 99.0%, ±2.0% | 100.0%, ±0.0% | 99.5%, ±1.0% | 100.0%, ±0.0% | — | — | 85.5%, ±4.9% | 95.0%, ±4.3% | 12.0%, ±6.4% | 56.0%, ±9.7% | 98.0%, ±2.7% | 100.0%, ±0.0% | 100.0%, ±0.0% | | |
| Adv. Train vs. Orig. Attacks | 94.6%, ±2.0% | 100%, ±0.0% | 53.2%, ±4.4% | 96.6%, ±1.6% | — | — | 49.4%, ±4.4% | 87.8%, ±2.9% | 12.4%, ±2.9% | 35.6%, ±4.2% | 92.4%, ±2.3% | 53.4%, ±4.4% | 96.8%, ±1.5% | — | — |
| BP vs. Orig. Attacks | 16.0%, ±7.2% | 1.0%, ±2.0% | 6.0%, ±4.7% | 6.0%, ±4.7% | — | — | 42.0%, ±9.7% | 55.0%, ±9.8% | 14.0%, ±6.8% | 64.0%, ±9.4% | 9.0%, ±5.6% | 6.0%, ±4.7% | 6.0%, ±4.7% | — | — |
| ANE vs. Orig. Attacks | 12.0%, ±6.4% | 0.0%, ±0.0% | 5.0%, ±4.3% | 5.0%, ±4.3% | — | — | 15.0%, ±7.0% | 39.0%, ±9.6% | 12.4%, ±4.3% | 40.0%, ±9.6% | 5.0%, ±4.3% | 5.0%, ±4.3% | 5.0%, ±4.3% | — | — |
| BP vs. NOTA Attacks | **99.0%**, ±2.0% | 1.0%, ±2.0% | 9.0%, ±2.5% | 9.2%, ±2.5% | **18.8%** ±3.4% | **91.8%** ±2.4% | 91.0%, ±5.6% | **100.0%**, ±0.0% | 15.0%, ±7.0% | 63.0%, ±9.5% | 13.0%, ±6.6% | **93.4%**, ±2.2% | **99.6%**, ±0.6% | **92.8%**, ±2.3% | **99.6%**, ±0.6% |
| ANE vs. NOTA Attacks | 9.0%, ±5.6% | 5.0%, ±4.3% | 8.6%, ±2.5% | 9.4%, ±2.6% | **46.8%** ±4.4% | **74.4%** ±3.8% | 55.0%, ±9.8% | **91.0%**, ±5.6% | 5.0%, ±4.3% | 37.0%, ±9.5% | 9.0%, ±5.6% | **52.6%**, ±3.4% | **95.4%**, ±1.8% | **54.0%**, ±4.4% | **95.0%**, ±1.9% |

The blanks represent original attacks, for which ANTI-NOTA (AN) loss does not exist as it is inherently adapted for NOTA.

Table 3: Attack Success Rates (ASRs), with 95% binomial confidence intervals, for Original and NOTA Attacks for CIFAR-100.

| Models | C&W Suite | | APGD CE | | APGD AN | | APGD DLR | | Square Attk | | DF | AA, Untrgtd | | AA-AN, Untrgtd | |
|---|---|---|---|---|---|---|---|---|---|---|---|---|---|---|---|
| | $L_2$, Conf:0 | $L_\infty$, Conf:0 | $L_2$, $\epsilon$:0.5 | $L_\infty$, $\epsilon$:0.031 | $L_2$, $\epsilon$:0.5 | $L_\infty$, $\epsilon$:0.031 | $L_2$, $\epsilon$:0.5 | $L_\infty$, $\epsilon$:0.031 | $L_2$, $\epsilon$:0.5 | $L_\infty$, $\epsilon$:0.031 | $L_2$, $\epsilon$:0.5 | $L_2$, $\epsilon$:0.5 | $L_\infty$, $\epsilon$:0.031 | $L_2$, $\epsilon$:0.5 | $L_\infty$, $\epsilon$:0.031 |
| No Defense | 98.3%, ±1.0% | 100.0%, ±0.0% | 98.8%, ±0.3% | 100.0%, ±0.0% | — | — | 99.7%, ±0.5% | 100.0%, ±0.0% | 41.3%, ±3.9% | 87.3%, ±2.7% | 83.5%, ±3.0% | 99.8%, ±0.3% | 100%, ±0.0% | — | — |
| Adv. Train vs. Orig. Attacks | 95.6%, ±1.8% | 100%, ±0.0% | 78.6%, ±3.6% | 97.2%, ±1.5% | — | — | 77.0%, ±3.7% | 96.0%, ±1.7% | 32.4%, ±4.1% | 62.2%, ±4.3% | 89.0%, ±2.7% | 78.8%, ±3.6% | 97.2%, ±1.5% | — | — |
| BP vs. Orig. Attacks | 51.8%, ±4.4% | 2.2%, ±1.3% | 33.2%, ±4.1% | 33.2%, ±4.1% | — | — | 78.4%, ±3.6% | 69.4%, ±4.0% | 47.6%, ±4.4% | 36.8%, ±4.2% | 24.2%, ±3.8% | 33.2%, ±4.1% | 33.2%, ±4.1% | — | — |
| ANE vs. Orig. Attacks | 40.2%, ±4.3% | 0%, ±0.0% | 31.2%, ±4.1% | 31.2%, ±4.1% | — | — | 51.8%, ±4.4% | 54.6%, ±4.4% | 31.2%, ±4.1% | 66.0%, ±4.2% | 16%, ±3.2% | 31.2%, ±4.1% | 31.2%, ±4.1% | — | — |
| BP vs. NOTA Attacks | 54.4%, ±4.4% | **34.6%**, ±4.2% | 33.2%, ±4.1% | 33.2%, ±4.1% | **58.2%** ±4.3% | **98%**, ±1.2% | **99.6%**, ±0.6% | **99.4%**, ±0.7% | 48.2%, ±4.4% | **49.2%**, ±4.4% | 32.8%, ±4.1% | **97.8%**, ±1.3% | **98.8%**, ±1.0% | **98.4%**, ±1.1% | **99.4%**, ±0.7% |
| ANE vs. NOTA Attacks | **49.4%**, ±4.4% | **30.3%**, ±3.7% | 31.2%, ±4.1% | 31.2%, ±4.1% | **83.4%** ±2.6% | **96.6%**, ±1.6% | 89.8%, ±2.7% | 90.4%, ±2.6% | 31.3%, ±3.7% | 68.0%, ±4.1% | 20.6%, ±3.0% | **89.2%**, ±2.7% | **96.6%**, ±1.6% | **94.6%**, ±2.0% | **98.6%**, ±1.0% |

The blanks represent original attacks, for which ANTI-NOTA (AN) loss does not exist as it is inherently adapted for NOTA.

#### 4.2.1.1 Unadapted Attacks vs. Defenses

With unadapted attacks, we find that both BP and ANE appear highly robust, as can be seen in Tables 2 and 3. Even strong baselines like AutoAttack show low ASR at or below natural error, and several other attacks show substantial reductions as well. Overall, considering the substantially successful defense against these benchmark attacks, NOTA defenses would seem an impressive advance in favor of increased adversarial robustness in classification systems. However, these results stem from unadapted attacks treating NOTA as a regular class, meaning that they often stop once they cross the true class boundary–even if they result in a NOTA classification. This also explains the apparent anomaly of how Square Attack appears more effective on its own than AutoAttack (which includes Square Attack as a component). Once an earlier component attack reaches NOTA, the attack pipeline stops rather than continuing to a non-NOTA misclassification. To sum up, these unadapted results overstate robustness.

#### 4.2.1.2    NOTA-Aware Adaptive Attacks vs. Defenses

Adapting attacks to the NOTA paradigm largely restore high ASR against both NOTA-defended models (Tables 2 and 3). The main exceptions are APGD-CE, CW $L_\infty$, and DeepFool, where robustness remains closer to the unadapted baseline. By contrast, APGD-DLR and our APGD-AN consistently drive ASR up, demostrating the PGD-based NOTA training may not protect against other PGD-based attacks that use a different loss function. Importantly, no evaluated attack version becomes weaker after being adapted.

For BP, NOTA-aware adaptations produce large ASR jumps, especially for APGD-DLR and AutoAttack. CW $L_\infty$ and APGD-CE show limited gains–consistent with BP's CE-driven training–while SquareAttack and DeepFool change little. BP's apparent robustness under unadapted evaluation evaporates once target selection and stopping criteria exclude the NOTA "honeypot".

ANE looks strong against unadapted attacks, but is similarly vulnerable under NOTA-aware evaluation. On both CIFAR-10 and CIFAR-100, APGD-DLR, APGD-AN, and AutoAttack regain high ASR. CW $L_2$ is a partial exception (weaker on CIFAR-10, stronger on CIFAR-100). As with BP, SquareAttack and DeepFool remain close to their original behavior. Notably, ANE retains some CE-specific resilience, which makes sense due to CE being used to generate NOTA examples during training.

#### 4.2.1.3    Comparing Adversarial Training with Adversarial NOTA Envelopment

With the advent of effective adapted attacks against the best NOTA defenses, a far more balanced comparison can be made between the standard for adversarial defense, adversarial training, and this complement of its defense paradigm, NOTA. First and foremost, our testing confirms that neither is a solution to the problem of adversarial attacks alone. Nevertheless, the comparison apparent in tables 2 and 3 are still of interest. Overall, there are some mixed performances reported here, with most, though not all, resulting in adversarial training (AT) providing less robustness against standard attacks than ANE provides against NOTA-aware adapted versions of those attacks. All results considered, ANE would appear to confer modestly more general robustness to a model than adversarial training, however, considering the results from the adapted NOTA-Aware attacks, both collapse and fail to defend models from evasion attacks overall, with several attacks achieving 90% or greater attack success rates on both.

### 4.2.2    IMDB

In Table 4, the clean accuracies and ASRs for the LLM-based classifier with and without the NOTA A2T defense are reported. While the NOTA defense does decrease clean test set accuracy from 93.66% to 87.75%, it even more dramatically reduces the ASR of GBDA from 98.40% to 31.79%. 31.79% demonstrates a very effective defense, considering that the natural error rate of the NOTA-defended model is 12.25%. When GBDA is adapted for NOTA, however, the ASR becomes 100.00%, demonstrating that NOTA-defended, LLM-based classifiers are still vulnerable to appropriately adapted adversarial evasion attacks.

Table 4: Test Clean Accuracies and Attack Success Rates (ASRs), with 95% binomial confidence intervals, for IMDB.

| | IMBD | | |
|---|---|---|---|
| Model | Clean Accuracy | GBDA ASR | NOTA GBDA ASR |
| No Defense | 93.66% | 98.40% $\pm$ 1.10% | — |
| NOTA A2T | 87.75% | 31.79% $\pm$ 4.08% | 100.00% $\pm$ 0.00% |

## 5    Discussion

In this paper, we begin by discussing and evaluating a group of open-set adversarial defense approaches which employ a none-of-the-above class to defend deep neural networks against evasion attacks. In investigating why this genre of defense, on its surface, is effective against many attacks, we discover common attack failure

modes. Particularly, we find that attacks often fail because: (1) they consider an adversarial attack that changes the predicted label to NOTA as a success and (2) they can incentivize making the NOTA class have the highest predicted probability. We solved (1) by defining a successful adversarial attack as one that leads to a predicted class that is neither the true class nor NOTA. This failure often occurs when defining stopping criteria. To address (2), we adjusted attacks that use "target selection" to not select the NOTA class. If "target selection" is not used, we investigated changing the loss. Using these, we modified eight prominent and highly effective benchmark attacks. In many cases, this largely recovered attack potency against NOTA defenses for both computer vision and NLP tasks.

We observe that our adapted attacks clearly show that present NOTA defenses are not sufficient to solve the core challenge of defending DNNs against evasion attacks. However, NOTA defenses can confer some small residual resilience to some adapted attacks that at least rivals adversarial training for the analyzed computer vision datasets. With several adapted attacks recovering ASRs back to 90% and above for both the evaluated computer vision and NLP tasks, we advise that future evaluations of NOTA-type or open-set-inspired defenses must begin by testing with NOTA-aware attacks. To this end, we make the NOTA-Aware adaptations to attacks created in this paper available for public use. Finally, if a NOTA-aware version of an attack is not available, the common failures and adaptations discussed in this paper could help practitioners adapt new attacks.

While these findings are critical, the evaluations of NOTA defenses and attacks could be improved in several ways. In this paper, the NOTA class is populated using methods based on common adversarial training approaches (Madry et al., 2017; Yoo & Qi, 2021). However, other promising methods exist for creating NOTA examples. One set of such methods are open-set training approaches (Ge et al., 2017; Chen et al., 2021). Also, other LLM adversarial attacks, could be adapted for NOTA defense, such as Geisler et al. (2024). Additionally, evaluations could be enhanced by using higher-resolution image classification datasets and more text tasks, such as Yelp Reviews (Zhang et al., 2015), AG News (Zhang et al., 2015), and LLM jailbreaking (Geisler et al., 2024).

### Acknowledgments

The views expressed in this article are those of the authors and do not reflect the official policy or position of the U.S. Naval Academy, the Naval Postgraduate School, Department of the Navy, the Department of Defense, or the U.S. Government. This work is funded in part by National Science Foundation under Grants nos. 2040209, and 2429835.

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
