# OpenReview forum: "Overcoming Open-Set Approaches to Adversarial Defense"
_TMLR — Accepted by TMLR_

### Review · Reviewer_fbSB · 2025-09-25

**Summary Of Contributions:**

Modern DNN models are vulnerable to adversarial examples and evasion attacks. A defense method creating None-of-the-Above (NOTA) class is proposed to defend against such attacks. Authors conducted experiments to show that existing NOTA solutions cannot completely resolve the challenges existing in evasion attacks. Authors give a clear introduction about the evasion attacks studied and carried certain number of experiments to support purposed argument.

**Audience:**

Yes

**Audience Explanation:**

Some individuals would be interested in knowing the findings of the paper as the paper studied on a popular question in terms of evasion attacks of DNN. The paper provides the idea that previous promising solutions for NOTA based defense still cannot resolve the core issues and makes the later researchers think on how they can effectively evaluate NOTA related solutions.

**Broader Impact Concerns:**

The paper does not have particular further concerns regarding Boarder Impact.

**Claims And Evidence:**

Yes

**Claims Explanation:**

In section 4, the authors presented results over CIFAR10 and CIFAR100 to show that NOTA based systems can effectively defend against certain attacks. On the other hand, they also showed that some core issues are not completely resolved and evasion attacks can still pose a threat to the scenario.

However, it is suggested that since the paper heavily depends on empirical observation. Experiments with other datasets such as ImageNet and other models are needed. Apart from that, tasks apart from image classification should also be studied.

**Requested Changes:**

- Some sections are redundant. For example, the introduction of classification models in DNN.
- Can authors elaborate in the section: threat model about in what case an attack can conduct PGD and AutoPGD, etc. and make the classification task fail?
- "Researchers Moosavi-Dezfooli, Fawzi and Frossard": please use \citep, \citet, \citeauthor or similar commands instead of directly writing authors names to do such reference.
- What is $\beta$ is section 3.2?
- Equation should be numbered.
- In section 3.3, why each attack needs to be proposed with a solution adapting to NOTA. Has anyone previously tried that?
- Apart from CIFAR10 and CIFAR100 and WRN, other models and datasets should also be taken into consideration. Also, for classification tasks, NLP tasks are suggested apart from image classification.

---

> ### Author Response · Authors · 2025-10-13
>
> We thank the reviewer for their kind words and suggested changes. We think that addressing the suggestions has improved the paper. Specifically, we:
>
> 1. Have modified the introduction of classification DNNs to include text inputs. A detailed discussion of DNN classifiers is necessary to link together the classification model and the various attacks and defenses.
>
> 2. Have added a sentence discussing that we use a whitebox threat model, which assumes that the gradients of a model, which are used by PGD methods, can be computed.
>
> 3. Now use \citet or \citep.
>
> 4. Now highlight that Beta is a random variable that is randomly sampled from a uniform distribution of the set [0,1] in Section 3.2. It is used to randomly select whether to push the adversarial example away from the true class or away from the NOTA class.
>
> 5. Now number the equations.
>
> 6. Edited the end of the introduction and Section 3.1.1 to remove confusing language regarding previous work from the literature, which made it ambiguous whether others had previously made these observations about NOTA defenses. We go into detail about each standard attack so that we demonstrate the application of our proposed taxonomy to create NOTA adapted attacks.
>
> 7. Evaluated the generalizability of the findings. We did so by applying a NOTA defense, a standard evasion attack, and a NOTA-adapted attack to a large language model (LLM) classification neural network, specifically for the IMDB sentiment classification task. (A task used as a benchmark for adversarial text methods.) Our findings for this new evaluation qualitatively match those for the computer vision datasets (CIFAR-10 and CIFAR-100).

---

### Review · Reviewer_7CjZ · 2025-09-27

**Summary Of Contributions:**

This paper studies existing adversarial attacks for ML model and defense techniques; and reports that NOTA-defenses results in too successful.  It then analyzes why it is so and proposes a way to attack NOTA-defenses more effectively and realistically.

**Additional Comments:**

I am happy to see a revised version during this review process.

**Audience:**

Yes

**Audience Explanation:**

While it is a bit hard to gain clear insights in its current form, the study seems thorough and it should give an interesting finding to the community.

**Claims And Evidence:**

No

**Claims Explanation:**

The current form is not really well written and have the following strength and weakness (concerns).
However, I think the paper tries to do thorough study, so clarifying the texts should show its contribution better.

Strength: This paper is trying to do thorough study of existing work and experiments; as long as those are better structured and clarified, it may be an interesting contribution.

Major concern:

The main concern is that this paper uses large part of the paper for describing existing methods in text with some figures that may or may not help understand the context.  If this is to compare different existing methods to highlight their strength and weakness etc. it is better to use tables or, the authors may want to add clear figures that show illustrations of each method concisely.
Also, the authors’ contributions, existing work, their weakness etc. are somewhat entangled in the text (even though they itemize contributions in the introduction).  It is a bit hard from the introduction etc. to see what is the main theme of the paper.
Further, in page 8 for example, the authors introduce target selection, stopping criteria… but they also mention that those aspects have already been discussed in the past work; here again it is a bit hard to see what is the exact contribution of the authors.
Especially, section 3 and 4 are very hard to read...

**Requested Changes:**

1. Restructure the paper to clarify the contributions, existing work comparison etc.  Simple flow, if possible, is preferable (e.g. general overview of attack/defense, about NOTA (that may not be analyzed thoroughly in the past), contribution (discovering its weakness or effective attack?) and proposed approach?)

2. Related to the above; any problem raised in 3.1 and 3.2 should be mentioned earlier and 3. should focus on the authors’ contributions or discoveries.

3. For experiment section, please summarize what you do concisely (while putting setups in tables etc.) and state observations clearly.  Currently, existing approaches, authors’ thoughts, results etc. are mixed up and it is hard to parse what is going on at first sight.

Minor changes:

1. page2 bottom; arrow (map) for f(x) is strange
2. page3 explanation of evasion attacks is a bit ambiguous; “an input x’ can be crafted…” this statement itself does not imply x’ is close to x.  (I know you explain it just after, but this sentence itself is not well written)
3. page 4, D is taking one or two inputs?
4. page 5, what is xhat?  And what is f_{x_i}?

---

> ### Author Response · Authors · 2025-10-13
>
> We thank the reviewer for their suggested changes. We think that addressing the suggestions has improved the paper. Specifically, we:
>
> Major Changes
>
> 1. Edited the end of the introduction and Section 3.1.1 to remove confusing language regarding previous work from the literature, which made it ambiguous whether others had previously made these observations about NOTA defenses. They have not. This is a contribution of the paper.
>
> Additionally, we have added a figure (Figure 1) to illustrate the issues that NOTA poses to standard attacks.
>
> 2. Have focused Section 3 solely on our contributions and discoveries.
>
> 3. Have edited the paragraphs (Section 4.2) discussing the results to focus on the main takeaways.
>
> Minor Changes
>
> 1. Fixed the arrow symbol.
>
> 2.  Rewrote the section to now highlight that the adversarial perturbation should be “small.”
>
> 3. Have standardized the distance function definition to take in two inputs. We have also modified the CW loss to use the same notation as the other attack losses.
>
> 4. Changed both to be x’.

---

> > ### Comment · Reviewer_7CjZ · 2025-10-15
> > **Thank you for the response**
> >
> > Thank you for the response; this is to acknowledge receipt.  Will be back later.

---

> > ### Comment · Reviewer_7CjZ · 2025-10-27
> > **Thank you for the response**
> >
> > I read the manuscript again.  The aim of this work has been clarified a bit.
> > However, I still think that the current structure and flow of this paper make its main argument difficult to understand.
> > I think the main message is very simple: just make attack methods aware of the existence of NOTA class in order to fool NOTA defense and it works effectively.   (More general claim may be "when evaluating robustness of defense, you should assume that attack methods are designed to deal with such defense methods")
> > Then, background is not required to talk about many methods, but could be for mentioning general attack/defense trend and reported effectiveness of NOTA defense for example.
> > Also, partially due to the too short title for each (sub)section, the focus and the claim of section 3 seems unfocused.
> > 3 failure modes are important, but how did you find them the most important components?  And how general are they?  Can they be applied to ANY attack method?  Rather than putting many small details of existing approach, I would strongly recommend the authors to focus on those scientific procedures.
> > So, after the background, with the research question "Is the reported effectiveness of NOTA reliable?  Is it only because the attack methods do not assume NOTA class?" and the main section should discuss logical steps to answer to this question and to find how to adjust the attack methods to fool it.  (This is just an example though)
> >
> > Minor concerns:
> > 1. page 3: embedding vector e_i in R^m   this m should not be bold
> > 2. why y_1,y_2,...,y_m?   c distinct classes?  c or m?
> > 3. Section 2; if the math is not really relevant to the main contribution, it would be better not to introduce many notations for just explaining existing method (what is z_y etc.?  not only "each z" but those should be defined clearly)
> > 4. what is y_t?
> > 5. for PGD, is it only for projecting to [0,1]^hwl?  only for image?
> > 6. also why for autoattach [0,1]^d?    avoid introducing those tiny maths if they are not really important
> > 7. in page 8   is between 0.05 and 0.95 really important to be mentioned here?  putting random details of existing work may confuse readers.
> >
> > I am sorry that I took some time for responding to the response; but I would still think the current form is not very well-written.  The main message (if it is something I mentioned above) is simple but is interesting if stated with more rigorous way of writing.
> >
> > (If this paper is accepted, I strongly recommend authors to reconsider the structure of the paper.)

---

> ### Author Response · Authors · 2025-10-27
> **Thank you for the Review.**
>
> Thank you for your review. While we contemplate how to address your main concern, we would like to address the minor concerns.
>
> 1. You are correct. We have unbolded m.
>
> 2. You are correct. We have modified this to use c.
>
> 3. We discuss the math because that is how these attacks are implemented and what we will be referencing when discussing how we applied our taxonomy to adapt each attack. $z_y$ is the logit associated with the correct label, an argument of the loss.
>
> 4. The sentence using $z_t$ has been removed.
>
> 5. Assuming that inputs are 0-1 max scaled, which is common for imaging data, is mentioned because that is the way these attacks are discussed in the referenced papers and how they are commonly implemented in libraries, such as in TorchAttack.
>
> 6. Same as in 5 above.
>
> 7. We add those details because we use those values when implementing Boundary Padding. We could move those details to the Experimental Setup subsection.

---

> > ### Comment · Reviewer_7CjZ · 2025-10-28
> > **Thank you for the response**
> >
> > Thank you for the response again.
> >
> > So, is the authors' approach specific to each attack method?
> > If there is any general scientific procedures or systematic approach, the paper may need to focus on that.
> > And each detail should be in experimental section or abstract in order to clarify the main claim in the main text.
> > But if it is really a case-study, then it should clearly state so in the paper; and in this case, each method should be more detailed with rigorous math instead actually.
> > If this is a ``discovery'' paper, then it would be better to focus on this aspect too (after introducing existing reports that say NOTA defense is effective, and analyze them scientifically, which led to this discovery etc etc.)
> >
> > I think the main message is interesting; but I still believe the current way of writing somehow confuses me a lot...

---

> > > ### Author Response · Authors · 2025-11-02
> > > **Proposed Changes**
> > >
> > > We thank the reviewer for their assistance in making the claims and contributions of the paper clearer. Would doing the following address the reviewer's concerns?
> > >
> > > 1. We change the main claim to "We analyze seven prominent adversarial evasion attacks developed for computer vision classification and one attack developed for natural language processing classification, identifying how these attacks fail in the presence of a NOTA defense. We use this knowledge to adapt these attacks and provide empirical evidence that adding a NOTA class alone does not solve the core challenge of defending DNNs against evasion attacks."
> > >
> > > 2. We move the detailed discussion of the standard evasion attacks to the methods section and discuss the attack details in the context of identifying failures in the presence of NOTA. Also, we discuss our adapted versions of the attacks in a more mathematical way, matching the discussion of the standard attacks. This leaves the Background section to focus on DNNs, the general idea of evasion attacks, the general idea of adversarial training, and the idea of NOTA defenses.

---

> > > > ### Comment · Reviewer_7CjZ · 2025-11-03
> > > > **Thank you again for the response**
> > > >
> > > > Thank you again for the response.
> > > > Yes, if this paper is about the case studies on those seven evasion attacks rather than proposing general systematic procedures, I agree that those change should make the claim more clear and scientifically reasonable.
> > > > As the authors point out, in this case, instead of removing the tiny mathematical details as I mentioned, it is better to make them more self-contained with sufficient math.  Then it should be clear from the mechanism of attacks and how the authors modify them that this approach is reasonable.  Readers may want to know more about the details on how this modification affect the behaviors of attacks (with figures, math) which will bring more insights (i.e., not only the resulting effectiveness of attacks but how the intermediate behaviors change, and if this is expected from the detailed math etc.).
> > > > Again, I think the main claim seems very interesting!

---

> > > > > ### Author Response · Authors · 2025-11-05
> > > > > **Proposed Changes Implemented**
> > > > >
> > > > > We thank the reviewer for all of the feedback. We think that this process has greatly improved the paper. We have made the changes to the main claim, adjusting every section of the paper to align with this shift. Notably, the math for the attacks were moved to the Methods section, combined with the NOTA attack adaptations, and edited to be more consistent across the various attacks.

---

> > > > > > ### Comment · Reviewer_7CjZ · 2025-11-05
> > > > > > **Thank you for the revision**
> > > > > >
> > > > > > Thank you for the revision!
> > > > > > It may take a few days for me to get back.

---

> > > > > > > ### Comment · Reviewer_7CjZ · 2025-11-07
> > > > > > > **I read the manuscript**
> > > > > > >
> > > > > > > Thank you for the revision; I think that the current updated version seems solid.

---

### Review · Reviewer_HvvB · 2025-09-29

**Summary Of Contributions:**

In this work, the authors provide several contributions: a clear list of failures of existing evaluations in existing attack evaluations (target selection, stopping criteria, objective function), they propose a complete benchmark with 7 distinct attacks, some of which are specifically tailored to open-set defense approaches, and experiments on CIFAR-10/100.

**Audience:**

Yes

**Audience Explanation:**

The topic is very relevant both from a theoretical perspective and for application. The presentation is very didactic and effectively reviews the domain.

**Broader Impact Concerns:**

no ethical concern

**Claims And Evidence:**

Yes

**Claims Explanation:**

The article is very clear and well-structured. The limitations outlined by the authors in existing evaluations are very well explained and illustrated in the experiments.

**Requested Changes:**

It would be interesting to include other open-set methods (e.g., G-OpenMax, ARPL) or hybrid defenses (e.g., OpenSet-AT) to show the generalizability of the findings. On the same note, cifar-10 and 100 are low-resolution datasets. It would be interesting to see if this is also true for high-resolution datasets, such as Imagenet (or perhaps a suggestion for future work).

---

> ### Author Response · Authors · 2025-10-13
>
> We thank the reviewer for their kind words and suggested changes. We think that addressing the suggestions has improved the paper. Specifically, we:
>
> - Evaluated the generalizability of the findings. We did so by applying a NOTA defense, a standard evasion attack, and a NOTA-adapted attack to a large language model (LLM) classification neural network, specifically for the IMDB sentiment classification task. (A task used as a benchmark for adversarial text methods.) Our findings for this new evaluation qualitatively match those for the computer vision datasets (CIFAR-10 and CIFAR-100).
>
> - Agree that using open-set techniques to populate the NOTA class and evaluating on higher-resolution image tasks are of great interest. We have added these as limitations of the current study and areas for future work at the end of the Conclusion section. Additionally, we discuss how the LLM-based analyses could be furthered.

---

### Author Response · Authors · 2025-10-16
**Update on NLP Classification ASR Results to tighten Confidence Intervals and slight results section restructure.**

In the update we provide today, there is no material change to the paper's writing or analysis from those provided Monday. We make the following minor changes:
1. We tighten confidence intervals on the recently added NLP sentiment analysis task.  Additional time allowed us to collect more samples and considerably tighten our or confidence intervals for the attack success rates in the IMDB results section of the paper. The confidence intervals shrink within the previous CI, and therefore do not change the overall analysis.
2. We make a minor structure edit to the results sections to clearly demarcate between CV and NLP results.

Thank you for your continued review and feedback.

---

### Author Response · Authors · 2025-11-07
**Minor grammar and math notation fix**

Today's edit corrects a few minor grammar and math notation inconsistencies only from the last edit.

---

### Decision · Action_Editor_sVXx · 2025-11-17

**Recommendation:** Accept as is

**Audience:**

Yes

**Audience Explanation:**

A reasonable portion of the community is interested in adversarial  attacks and defense mechanisms.

**Claims And Evidence:**

Yes

**Claims Explanation:**

The paper studies a defense strategy for machine learning models that adds a "none-of-the-above" class to reject unknown or adversarial inputs. The paper studies the effectiveness of this defense mechanism. They show that some existing attacks fail simply because they are not designed to account for this additional class. By investigating different computer vision and NLP tasks/attacks, the paper provide empirical evidence that these defenses can be easily bypassed, concluding that the method alone does not necessarily solve the core issue.

The paper's review process was highly interactive.   Initially, some reviewers found the paper poorly structured and made some suggsestions to improve the presentation of the paper. In response to this and other feedback, the authors performed two major revisions. They expanded their experiments and undertook a major restructuring of the paper to clarify its main claims. The reviewers found the revised version above the acceptance threshold and found the paper a solid contribution to the ML community.